# Predicting the Performance of Functional Materials Composed of Polymeric Multicomponent Systems Using Artificial Intelligence—Formulations of Cleansing Foams as an Example

**DOI:** 10.3390/polym15214216

**Published:** 2023-10-25

**Authors:** Masugu Hamaguchi, Hideki Miwake, Ryoichi Nakatake, Noriyoshi Arai

**Affiliations:** 1Department of Mechanical Engineering, Keio University, 3-14-1 Hiyoshi, Kohoku-ku, Yokohama 223-8522, Kanagawa, Japan; arai@mech.keio.ac.jp; 2Kirin Central Research Institute, Kirin Holdings, 26-1, Muraoka-Higashi 2-Chome, Fujisawa 251-8555, Kanagawa, Japan; 3Research Institute, Fancl Corporation, 12-13 Kamishinano, Totsuka-ku, Yokohama 244-0806, Kanagawa, Japan

**Keywords:** QSPR, AI, machine learning, cleansing capability, super-multicomponent system

## Abstract

Cleansing foam is a common multicomponent polymeric functional material. It contains ingredients in innumerable combinations, which makes formulation optimization challenging. In this study, we used artificial intelligence (AI) with machine learning to develop a cleansing capability prediction system that considers the effects of self-assembled structures and chemical properties of ingredients. Over 500 cleansing foam samples were prepared and tested. Molecular descriptors and Hansen solubility index were used to estimate the cleansing capabilities of each formulation set. We used five machine-learning models to predict the cleansing capability. In addition, we employed an in silico formulation by generating virtual formulations and predicting their cleansing capabilities using an established AI model. The achieved accuracy was *R*^2^ = 0.770. Our observations revealed that mixtures of cosmetic ingredients exhibit complex interactions, resulting in nonlinear behavior, which adds to the complexity of predicting cleansing performance. Nevertheless, accurate chemical property descriptors, along with the aid of in silico formulations, enabled the identification of potential ingredients. We anticipate that our system will efficiently predict the chemical properties of polymer-containing blends.

## 1. Introduction

Quantitative structure–property relationships (QSPRs) are used in several fields, such as environmental chemistry [1], drug design [2], and materials science [3], demonstrating their versatility and ability in scientific research and industrial applications. The QSPR is a statistical and mathematical method for expressing the relationship between chemical structure and physical properties, which enables the rapid prediction of physical properties. This expectation is particularly high in materials science, where recent advances in synthetic technology have enabled the creation of nearly an infinite variety of polymers. The materials produced by this synthesis require highly desirable physical properties. In recent years, several materials have been improved by adding different mixtures. For example, adding nanoparticles or nanofillers to a material to tailor a nanocomposite improves its thermal response and ionic conductivity [4,5,6,7,8]. Several attempts have been made to predict the physical and chemical properties of such complex systems using QSPRs. In the context of the prediction of physical or chemical properties of mixtures, reports predicting the flash point [9], diffusion coefficient [10], boiling point [11], refractive indices [12], or toxicity [13] of binary mixtures are available. Surfactants are representative functional polymers, and studies related to surfactants were performed that predicted the properties such as critical micelle concentration, cloud point, and the hydrophilic–lipophilic balance of binary mixtures [14,15]. From the perspective of multicomponent mixtures comprising three or more substances, examples of predicting properties such as vapor–liquid critical volume are available [16].

In several reports, molecular descriptors were acquired, and mixtures were characterized using methods such as weighted averages. Subsequently, predictive models are often constructed using linear prediction models (or their derivatives), machine learning, or deep-learning approaches. However, in the case of multicomponent systems, particularly with large-molecular-weight materials, challenges still exist for QSPRs regarding prediction accuracy [17], and reports using predictive models for estimating mixture effects are scarce. The lack of extensive research on predicting the properties of multicomponent mixtures, particularly those involving surfactants, and applying these predictive models to infer mixture effects and facilitate product design underscores the novelty and significance of this study.

The cleansing foam used in cosmetics is a typical polymeric multicomponent system, and polymers, such as polyethylene glycol or polyglycerin, are often used as ingredients. It is used to wash excess sebum and dirt from the skin, and a makeup remover is used to remove makeup cosmetics. In recent years, a growing need has been observed for highly functional cleansing foams because people want to use only one product for cleansing foam and makeup removal for reasons of time (shortening procedures) and ecology (saving water and reducing chemical emissions into the environment) [18]. Solvent-based cleansing agents, such as makeup remover oils, are highly soluble in makeup products, which contain oil and pigments, resulting in excellent removability. However, solvent-based cleansing agents are associated with problems such as high environmental impact, high material costs, and a feeling of residual oiliness after rinsing [19]. By contrast, surfactant-based cleansing agents such as cleansing foams have excellent rinsing properties but weak oil removability because they are primarily water-based. In this study, the latter approach was used to improve the cleansing performance of the foams.

Cleansing foams are composed of numerous components. They contain several types of surfactants, polyols, pH adjusters, and water, making optimizing the formulations difficult because an infinite number of ingredient combinations are possible. Therefore, artificial intelligence (AI) using machine learning has been introduced into formulation design to construct a cleansing capability prediction system that considers the effects of surfactant self-assembly and chemical characteristics of ingredients.

The focus of this study is twofold. First, it aims to extend our understanding of property prediction for multicomponent mixtures. Second, it aims to determine high-performance mixing conditions using predictive models. To achieve these goals, we employed various machine-learning methods. In addition, in silico simulations have been introduced to assist human formulators in achieving desirable products during product development.

## 2. Materials and Methods

### 2.1. Evaluation of Cleansing Capability

Cleaning foams consisting of 537 samples of ionic surfactants, amphoteric surfactants, nonionic surfactants, polyols, a pH adjuster, and water were prepared by thorough mixing and stirring. Examples of ingredients and formulations are listed in Table 1 and Table 2, respectively. Each sample comprised ~20% ionic surfactants, 10% nonionic surfactants, 10% polyols, 1% citric acid, and 60% water by weight. To study the prepared samples, a waterproof eyeliner pencil was placed on a piece of white artificial leather that was dried for 30 min. Then, 0.1 mL of the corresponding cleansing foam sample was added to the dried eyeliner, rubbed 30 times, rinsed, and dried. A schematic of all the procedures is shown in Figure 1.

The cleansing capability was evaluated using the eyeliner pencil residual ratio, which was calculated using the color differences as follows:(1)$$\mathrm{Cleansing}\;\mathrm{capability}\;(\%)=\frac{\sqrt{{\left(L_{1}^{*}-L_{2}^{*}\right)}^{2}+{\left(a_{1}^{*}-a_{2}^{*}\right)}^{2}+({b_{1}^{*}-b_{2}^{*})}^{2}}}{\sqrt{{\left(L_{1}^{*}-L_{0}^{*}\right)}^{2}+{\left(a_{1}^{*}-a_{0}^{*}\right)}^{2}+({b_{1}^{*}-b_{0}^{*})}^{2}}}*100$$where *L** indicates lightness, and *a** and *b** indicate chromaticity. (*L**, *a**, *b**) represents the color space value measured using a colorimeter (CM-2600d, Konica Minolta, Inc., Tokyo, Japan). (*L**_0_, *a**_0_, *b**_0_*)*, (*L**_1_, *a**_1_, *b**_1_*)*, and (*L**_2_, *a**_2_, *b**_2_) represent the color space values of the white artificial leather before applying the eyeliner pencil, after applying it, and after cleaning it, respectively [20].

### 2.2. Modeling of AI

#### 2.2.1. Data Processing

We trained the AI on the prescribing data and modeled them using descriptors and Hansen dissolution parameters to incorporate chemical information. Figure 2 shows the data-processing flowchart.

#### 2.2.2. Molecular Descriptors

A molecular descriptor is a numerical molecular property extracted from a chemical structure. Each type of molecular descriptor is related to a specific type of interaction between chemical groups in a particular molecule. Descriptors are used to predict the chemical properties of not only single chemicals but also chemical mixtures. In addition, descriptors have been applied to predict the critical micelle concentration (CMC) of gemini surfactants [21,22]. Therefore, we extracted information from ingredients and predicted the cleansing capabilities of the prepared formulations using molecular descriptors. The structural formula of each ingredient was determined using ChemDraw and converted into a Simplified Molecular Input Line Entry System (SMILES). Regarding the specification of the degree of polymerization, we adopted the representative degree of polymerization for each raw material. Descriptor values were then calculated from the SMILES of each ingredient using the chemoinformatic tools rdkit [23] and PaDEL-descriptor [24]. Entries with infinite or only one value were removed, and a *k*-NN imputer was applied to predict missing values. The weighted average of each ingredient was then calculated using the molar or weight fraction to estimate the descriptor values of the mixture ingredients. 

#### 2.2.3. Hansen Solubility Parameters

We applied Hansen solubility parameters (HSPs) to predict the cleansing capability. The HSPs were developed by Hansen to predict the ability of a material to dissolve in another material, forming a solution. The HSP distance between the solute and solvent is generally calculated to estimate whether a solute dissolves in a solvent. Some studies have used the HSPs to predict the properties of surfactants [25,26]. In this study, instead of the solute and solvent, we calculated the distance between each sample and obtained the cleansing samples with the highest cleansing capability. We adopted this procedure because the solute, the eyeliner in this study, was made of several ingredients, and it was difficult to identify its structural formula. The HSP distance was defined as {4*(dD_1_-dD_2_)^2^ + (dP_1_-dP_2_)^2^ + (dH_1_-dH_2_)^2^}^0.5^, where dD_1_, dP_1_, and dH_1_ are the values of each sample—the average calculated based on each component’s proportion by wight in the mixture—and dD_2_, dP_2_ and dH_2_ are the average values of the three highest cleansing capabilities in our samples. The HSPs would better estimate the effects of the interactions between the ingredients in a formulation than the descriptor method, in which measuring the nonlinear effect of the ingredient interactions is challenging. The HSP values were calculated using the Hansen Solubility Parameters in Practice (HSPiP) software of version 5.0.09. Because some HSPs cannot be calculated using HSPiP for molecules with high molecular weights, missing HSP values were imputed using a *k*-NN imputer for the descriptor calculation. 

#### 2.2.4. Modeling and Feature Selection

The number of explanatory variables was >1000 when descriptors and HSPs were used. Therefore, we applied machine learning to obtain laws to predict cleansing performance based on these numerous features. Three types of machine-learning algorithms are available: supervised learning, unsupervised learning, and reinforcement learning. Supervised learning (regression) was chosen for this study to predict the results within a continuous output. The input dataset is described in Section 2.1 and Section 2.2, and the output is the cleansing capability. In this study, we aimed to capture the inherent behavior of surfactants in cleansing forms, anticipating their utilization in more generic applications, such as predicting properties other than cleansing capability. Therefore, we did not focus on developing prediction models specialized for cleansing capability; instead, we adopted representative machine-learning models. We adopted two decision-tree-based models (random forest and extra tree regressors), two linear-based models (lasso and partial least squares), and one support-vector-machine-based model (support vector regressor). The hyperparameters are listed in Table 3. Each model exhibits unique characteristics: decision-tree-based models help capture nonlinear relationships in the data, and linear models are particularly suitable when a linear relationship is assumed between variables. Support-vector-machine-based models are suitable for predicting high-dimensional data. We aimed to develop versatile models by employing these diverse methodologies and uncover new potential for understanding surfactant behavior in cleansing forms. The hyperparameters were optimized using a grid-search method. All explanatory features were standardized with a mean of zero and standard deviation of one. Because numerous features cause noise in the modeling, we adopted the Boruta method [27] to reduce the noise from unimportant features.

#### 2.2.5. Modeling Evaluation

Herein, we employed a machine-learning model and optimized its hyperparameters using grid-search cross-validation, which is a popular method for hyperparameter tuning that works systematically through multiple combinations of parameter tunings. The scoring metric used to evaluate the performance of the model was the coefficient of determination, denoted as *R*^2^, which represents the proportion of variance for a dependent variable that is explained by the independent variables. The *R*^2^ values were calculated as follows:(2)$$R^{2}=1-\frac{\sum_{1}^{n}{\left(y_{i}-\hat{y_{i}}\right)}^{2}}{\sum_{1}^{n}{\left(y_{i}-\bar{y}\right)}^{2}},$$
where $\hat{y_{i}}$, $y_{i}$, and $\bar{y}$ represent the predicted, actual, and mean values of the actual output, respectively. The dataset comprised 537 samples, and ten-fold cross-validation was applied to calculate the accuracy. For the computations, 90% of the data were allocated as training data, and the remaining 10% as test data. This computation was conducted ten times, ensuring that all data were used as test data at some point. The average value of the ten *R*^2^ scores of the foldout data was accepted as the model performance. The modeling was executed five times with different random seeds, which were applied to the modeling of tree models and cross-validation split, and the averaged values were calculated as a result of accuracy. 

### 2.3. In Silico Formulation

To evaluate whether the AI models could support human formulators, formulations were virtually created with a computer using the rules described below. We call this procedure the ‘in silico formulation’.All ingredients were assigned to one of six categories (the same categories described in Table 1): anionic surfactants, amphoteric surfactants, nonionic surfactants, polyols, a pH adjuster (only citric acid), and a base (only water).To compare the predicted and actual cleansing capabilities, the selection of anionic and amphoteric surfactants was restricted to one type: the anionic surfactant was restricted to potassium cocoyl glutamate, and amphoteric surfactant was restricted to lauramidopropyl hydroxysultaine.Only one ingredient was selected from each category; for example, two nonionic surfactants could not be selected for one formulation.The addition rates of each ingredient, except for the pH adjuster (citric acid) and water, were randomized for each category within the predefined ranges described in Table 4. The addition rate of citric acid was fixed with the value of 0.8 weight%, and the addition rate of water was calculated such that the sum of all the ingredients was 100%.In the procedure, 10^5^ formulations were made, which were predicted with the best model described in Section 2.2.4.

To validate the predictions made by the in silico formulation, the actual cleansing capabilities of some formulations were measured experimentally (the formulations of the measured samples are shown in the Section 3).

## 3. Results

### 3.1. Evaluation of AI Modeling

An AI model was established to predict the cleansing capability. The prediction accuracy of each model is listed in Table 5. The best prediction accuracy was obtained with the *R*^2^ value of 0.770. The prediction accuracy increased significantly with the use of the descriptors. The results of the best model, extra tree regressor, using molecular descriptors, Hansen solubility index, and feature extraction are shown in Figure 3. An *R*^2^ value of 0.770 translates to 15% when converted to a root-mean-square error (RMSE). We believe this accuracy level is sufficient for screening purposes, such as opting not to conduct low-predictive cleansing capability experiments before engaging in experiments using actual substances. Such preliminary filtering enables a more efficient allocation of resources to experiments with higher probabilities of success, thereby optimizing the overall research process.

### 3.2. In Silico Formulation and Actual Cleansing Capabilities

The cleansing capabilities of the in silico formulations are shown in Figure 4 and Figure 5. A box with light and dark gray color in these figures indicates the middle 50% of the data (that is, the middle two quartiles of the data distribution), and horizontal bars display all points within 1.5 times the interquartile range (that is, all points within 1.5 times the width of the adjoining box), or all points at the maximum or minimum extent of the data. As shown in Figure 4, eicosaglycerol hexacaprylate exhibited the highest cleansing capability at both the median value (middle horizontal line in each box) and best value (top horizontal line).

In Figure 5, the formulation data were stratified into two categories: those not using nonionic surfactants and those using nonionic surfactants. Next, each category was stratified into subcategories based on the polyols to estimate their interactions with nonionic surfactants and polyols. The addition of nonionic surfactants increased the cleansing capabilities, and hydrophobic PPG-9 diglyceryl ether or cyclohexylglycerin, which have lower inorganic and organic balance (IOB) values, boosted the cleansing capability more than glycerin.

Several formulations were selected to validate the predicted data obtained from the in silico formulation, and their cleansing capabilities were measured. The formulations and results are shown in Table 6 and Figure 6. Nonionic surfactants eicosaglycerol hexacaprylate and cyclohexylglycerin/PPG-9 diglyceryl ether, (A) and (B), showed the highest cleansing capabilities among the actual formulations. Formulations with other nonionic surfactants and cyclohexylglycerin/PPG-9, (C) and (D), showed lower cleansing capabilities. Formulations with glycerin, (E) and (F), showed much lower cleansing capabilities, regardless of the type of nonionic surfactant. These tendencies correspond to the results shown in Figure 4 and Figure 5.

## 4. Discussion

The use of descriptors increased the prediction accuracy of all the models, indicating that the chemical properties expressed as molecular descriptors successfully enabled the prediction of cleansing capabilities. HSPs improved the prediction accuracy for several models, but the accuracy was insufficient for models with descriptors, indicating that descriptors were more informative than HSPs.

Furthermore, weight% was suitable for linear-based models for the weighted average calculation, whereas no difference was observed for tree-based models. The mol% of the weighted average is potentially more accurate based on stoichiometry. However, because water constitutes > 97 mol% on average in the formulations owing to the high molecular weights of the surfactants, the influence of water was more dominant in the mol% calculation. Linear-based models were more affected by this influence than tree-based models. To predict cleansing capability, nonlinear behavior should also be considered owing to the interactions between surfactants and water molecules and their self-assembly. Tree-based models are typically more suitable for nonlinear predictions; therefore, their prediction accuracies are higher than those of linear-based models. The most accurate method is the elastic tree regressor. Because this method is based on decision tree models, its accuracy will decline when the data intended for inference fall into extrapolation regions relative to the training data. In such cases, additional experiments must be conducted to augment data and retrain the model. Although we employed descriptors to enhance the generalization capability of the model, it was estimated that the accuracy of predicting the cleansing capability of samples made with ingredients not present in the training data would be lower than the accuracy calculated in this study.

The raw ingredients used included polymer-based components such as PEG-20 glyceryl triisostearate; however, the prediction accuracy was maintained. In addition, as shown in Figure 4 and Table 6, the formulations using PEG-20 glyceryl triisostearate exhibited a lower cleansing performance than those using shorter-chain raw materials. This suggests that the length of the polymer may not be a key factor influencing the cleansing performance; instead, it is likely that the higher-order structure between the ingredients plays a more significant role in cleansing.

The in silico formulation helped us understand the effect not only of each material on cleansing capabilities but also of combinations of materials with the consequence of molecular interactions. The in silico formulation assisted in developing formulations with higher cleansing capabilities.

In this study, applying prediction and in silico formulation methods, we identified a cleansing foam formulation consisting of eicosaglycerol hexacaprylate and cyclohexylglycerin/PPG-9 that exhibited a high cleansing capability of >85% for the removal of waterproof eyeliners. 

## 5. Conclusions

Using AI with machine learning, we built QSPR models that incorporated super-multicomponent ingredients, including polymers, to estimate the effects of surfactant self-assembly and chemical characteristics of the ingredients. An accuracy of *R*^2^ = 0.770 (RMSE = 15%) was obtained to predict the cleansing performance, which was sufficient for ingredient screening. Nonlinear behavior, i.e., interactions among cosmetic ingredients in formulations, makes it more difficult for formulators to predict their performance. However, a high accuracy was obtained by incorporating chemical characteristics with descriptors. Based on the molecular structure of the ingredients and surfactant self-assembly, this AI prediction model showed higher accuracy than conventional approaches, such as multiple linear regression. Using in silico formulations, formulators can obtain information on which ingredients should be selected to achieve the highest cleansing capabilities. These findings suggest that cleansing performance is not merely dependent on the polymer length of the ingredients but also on the higher-order structures resulting from the interactions between the ingredients. This prediction model and in silico formulation significantly reduce the effort required for cosmetic development. In summary, we achieved the following results:A QSPR model was constructed for super-multicomponent ingredients, including polymers, achieving an accuracy of *R*^2^ = 0.770 (RMSE = 15%), sufficient for product development screening.Using an in silico formulation, we predicted the optimal combination of the ingredients.The application of these technologies reduces the developmental effort and optimizes the overall development process.

Herein, we demonstrated the applicability of QSPRs to multicomponent mixtures, focusing specifically on cleansing foams. Based on the insights provided by QSPRs, we successfully identified an optimal combination of ingredients suitable for product development. We believe this methodology has high generalizability, facilitating the discovery of ideal combinations with minimal experimentation in various fields, not limited to cosmetics but also in drug development or other material designs. Therefore, the QSPR-based approach can emerge as a potent tool, yielding significant benefits in these industries.

## Figures and Tables

**Figure 1 polymers-15-04216-f001:**
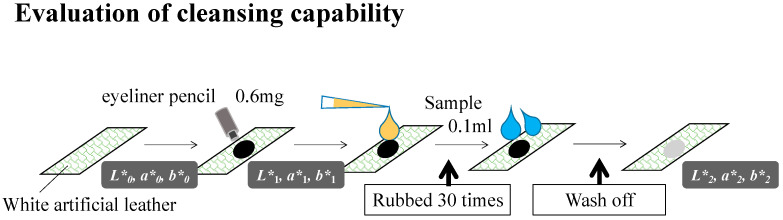
Schematic of the evaluation test to determine the cleansing capability of the prepared samples, Equation (1).

**Figure 2 polymers-15-04216-f002:**
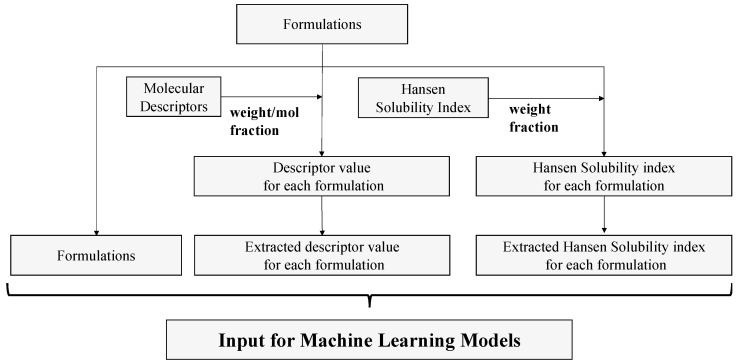
Data-processing flow.

**Figure 3 polymers-15-04216-f003:**
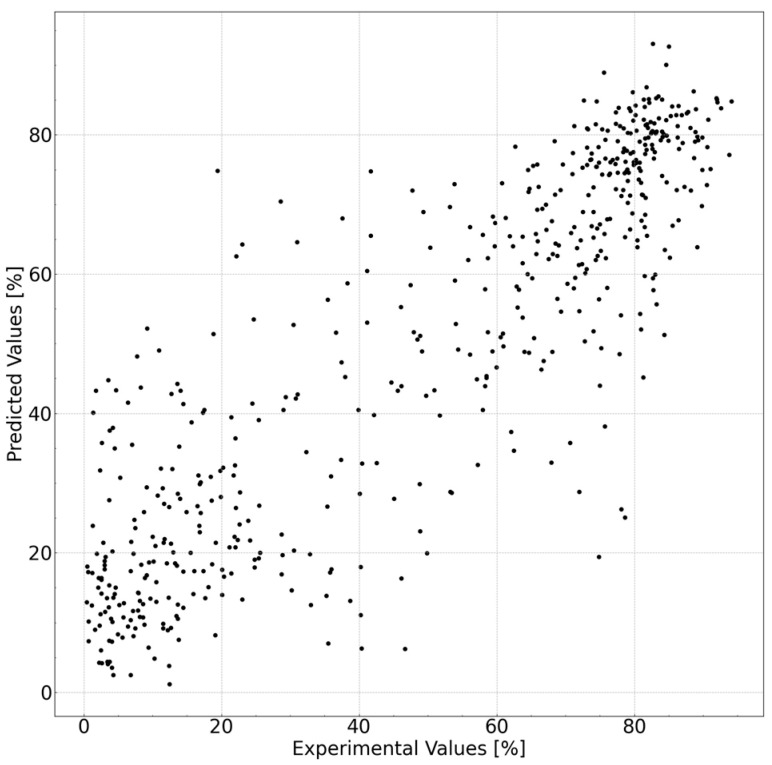
Experimental vs. predicted values of the cleansing capabilities of the cleansing foam formulations with the best model (10-fold cross validation was executed and all foldout data are shown in this plot).

**Figure 4 polymers-15-04216-f004:**
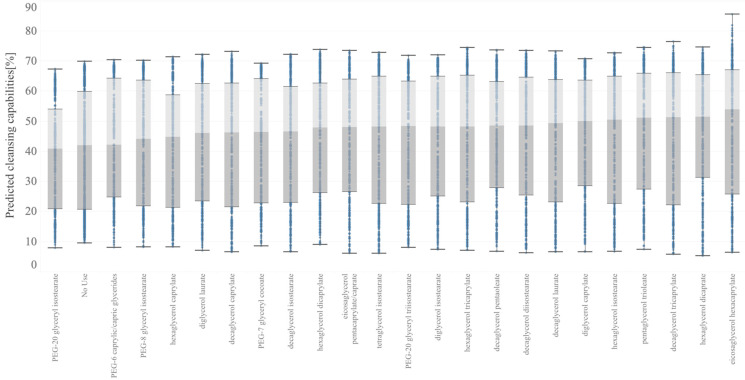
Box plots of cleansing capabilities with the in silico formulation based on the best model shown in Section 3.1. Predicted data are stratified with nonionic surfactants.

**Figure 5 polymers-15-04216-f005:**
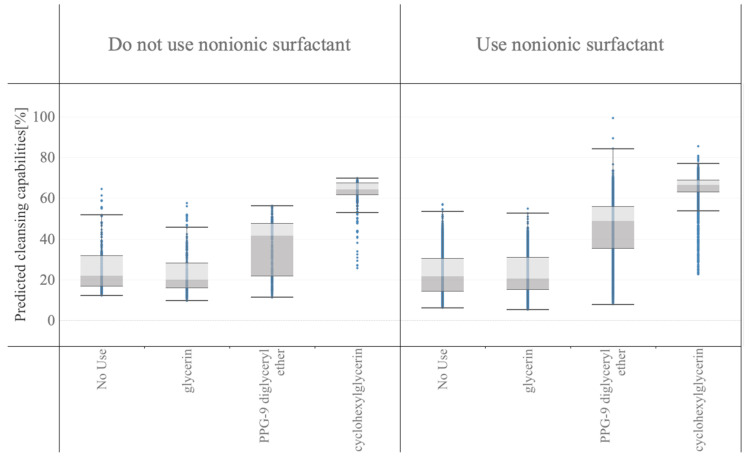
Box plots of predicted cleansing capabilities with the in silico formulation based on the best model shown in Section 3.1. Predicted data are stratified with nonionic surfactants and polyols.

**Figure 6 polymers-15-04216-f006:**
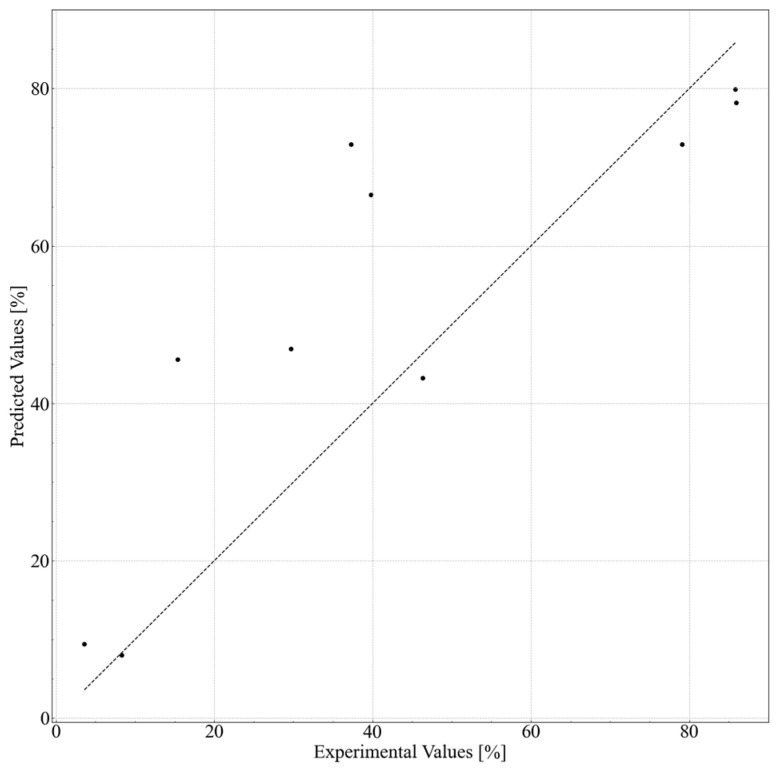
Experimental vs. predicted values of the cleansing capabilities for in silico formulations.

**Table 1 polymers-15-04216-t001:** Examples of ingredients used in the formulations prepared in this study.

Category	The Number of Ingredients	Ingredient Examples
Anionic surfactant	8	Potassium cocoyl glutamate
		Potassium cocoyl glycinate
Amphoteric surfactant	4	Lauramidopropyl hydroxysultaine
		Sodium cocoamphoacetate
Nonionic surfactant	24	Eicosaglycerol hexacaprylate
		Decaglycerol isostearate
		PEG-20 glyceryl triisostearate
Polyol	33	PPG-9 diglyceryl ether
		Cyclohexylglycerin
		Glycerin
pH adjuster	1	Citric acid
Base	1	Water

**Table 2 polymers-15-04216-t002:** Example of cleansing foam samples in this study.

Category	Material Name						
Anionic surfactant	Potassium cocoyl glutamate	6	7	13	8	8	7
Amphoteric surfactant	Lauramidopropyl Hydroxysultaine	5	9	4	9	7	7
Nonionic surfactants	Eicosaglycerol hexacaprylate	10	10			10	
	Decaglycerol isostearate			10			
	PEG-20 glyceryl triisostearate				10		10
Polyols	PPG-9 diglyceryl ether		11		11		
	Cyclohexylglycerin	9		14			
	Glycerin					11	13
pH adjuster	Citric acid	0.8	0.8	0.8	0.8	0.8	0.8
Base	Water	69.2	62.2	58.2	61.2	63.2	62.2

**Table 3 polymers-15-04216-t003:** Hyperparameter set for modeling.

Model Name	Hyperparameter	Values
Extra tree regressor	The number of trees	10, 50, 100, 500, 1000, 2000
	The number of features to consider when looking for the best split	(the number of features)^0.5^, log_2_(the number of features), the number of features
	The maximum depth of the tree	10, 20, 30
Random forest regressor	The number of trees	10, 50, 100, 500, 1000, 2000
	The number of features to consider when looking for the best split	sqrt, log2, none
	The maximum depth of the tree	10, 20, 30
Support vector regressor	C	0.1, 1, 10
	Epsilon	0.01, 0.1, 1
	Kernel	Linear, poly, rbf
Partial least squares regressor	The number of components	1 to 20 (integer)
Lasso regressor	Alpha	0.0001, 0.001, 0.01, 0.1, 10

**Table 4 polymers-15-04216-t004:** Condition of in silico formulation.

Category	Material Name	Randomize the Addition Rate	Minimum Weight% *	Maximum Weight% *
Anionic surfactant	Potassium cocoyl glutamate	Yes	3	20
Amphoteric surfactant	Lauramidopropyl hydroxysultaine	Yes	3	20
Nonionic surfactant	Eicosaglycerol hexacaplyrate PPG-20 glyceryl triisosterarate etc. (24 kinds in total)	Yes	0	10
Polyols	PPG-9 diglyceryl ether glycerin etc. (33 kinds in total)	Yes	0	30
pH adjuster	Citric acid	No	0.8% (fixed ratio)	
Base	Water	No	100-Σ (other material amount)	

* Water content was excluded in the weight% expression.

**Table 5 polymers-15-04216-t005:** Prediction accuracy of each model.

	Mol/Weight Fraction	Descriptors	Hansen Solubility Index	Feature Extraction	Extra Tree Regressor	Random Forest Regressor	SVR	Lasso	PLS
1	Weight	Not used	Not used	Not used	0.661	0.652	0.488	0.526	0.503
2	Mol	Not used	Not used	Not used	0.668	0.643	0.469	0.503	0.472
3	Weight	Used	Not used	Not used	0.751	0.715	0.567	0.565	0.541
4	Mol	Used	Not used	Not used	0.753	0.751	0.509	0.526	0.519
5	Weight	Not used	Used	Not used	0.725	0.657	0.320	0.582	0.535
6	Mol	Not used	Used	Not used	0.719	0.654	0.305	0.556	0.505
7	Weight	Used	Used	Not used	0.748	0.711	0.568	0.582	0.543
8	Mol	Used	Used	Not used	0.751	0.749	0.524	0.537	0.526
9	Weight	Used	Used	Used	0.770	0.738	0.671	0.564	0.532
10	Mol	Used	Used	Used	0.768	0.753	0.620	0.523	0.504

**Table 6 polymers-15-04216-t006:** Formulations for comparison of predicted and actual cleansing capabilities.

Category	Material Name	(A)	(B)	(C)	(D)	(E)	(F)
Anionic surfactant	Potassium cocoyl glutamate	6	7	13	8	8	7
Amphoteric surfactant	Lauramidopropyl Hydroxysultaine	5	9	4	9	7	7
Nonionic surfactants	Eicosaglycerol hexacaprylate	10	10			10	
	Decaglycerol isostearate			10			
	PEG-20 glyceryl triisostearate				10		10
Polyols	PPG-9 diglyceryl ether		11		11		
	Cyclohexylglycerin	9		14			
	Glycerin					11	13
pH adjuster	Citric acid	0.8	0.8	0.8	0.8	0.8	0.8
Base	Water	69.2	62.2	58.2	61.2	63.2	62.2
Cleansing capability test	Prediction/%	81.2	78.3	71.9	42.1	7.4	14.4
	Actual/%	85.8	85.9	39.8	46.3	8.3	3.6

## Data Availability

The data presented in this study are available on request from the corresponding author. However, some data are not publicly available because the database comprising individual formulation data is utilized for product development.

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
