# Peer review of "Predicting the Performance of Functional Materials Composed of Polymeric Multicomponent Systems Using Artificial Intelligence—Formulations of Cleansing Foams as an Example"

_polymers, 2023, doi:10.3390/polym15214216_

Round 1

Reviewer 1 Report

The authors developed a cleaning capacity prediction system using machine learning techniques. Much of the content does not meet the requirements for publication.

1. In recent years, machine learning algorithms have been very much used in the prediction of material properties. By summarising more literature, the application of machine learning in related fields is discussed in detail.

2. The authors seem to show only the prediction results of the machine learning model. But there is no detailed presentation of the dataset constructed, e.g. how many data samples? What is the number of samples in the training set and test set. A portion of the samples can be shown and it would be more beneficial to the field if an open source dataset is provided.

3. The authors chose 5 types of machine learning models and why they chose them needs to be discussed in detail.

4. The authors used R2 as an evaluation parameter for the prediction model.The value of R2 is only 0.770, which does not seem to be high. Please use multiple assessment parameters for a comprehensive assessment.

5. In the discussion section, the drawbacks and shortcomings of the proposed method should be demonstrated.

6. In the current version, the results presentation part is very insufficient, and it is suggested to add more visual comparison results between predicted and real results.

7. In this paper, only six already available methods have been used to predict the cleaning capacity. The analysis part is very insufficient to effectively illustrate the validity and feasibility of the proposed methods.

8. Many descriptions in the manuscript are not reasonable. For example, "The achieved accuracy was R2 = 0.770." The writing of the paper needs to be greatly improved.

The writing of the paper needs to be greatly improved.

Reviewer 2 Report

A clear in silico formulation usage study by generating virtual formulations and predicting the cleansing capability of cleansing foam by using an established AI model. It would have been useful to include gemini surfactants in the study.

Reviewer 3 Report

1. the abstract should be finalised. It is necessary to specify the scientific novelty.

2. The literature review is poorly done. More information is needed on the application of machine learning technologies.

3. it is necessary to describe in more detail the proposed machine learning methods for the research tasks.

4. Show an example of using the proposed technology on different application problems.

5. Give a summary table showing the advantages of the proposed method.

6. Improve the conclusions.

Moderate editing of English language required

Round 2

Reviewer 3 Report

 Accept in present form